# Artificial Intelligence-Powered Whole-Slide Image Analyzer Reveals a Distinctive Distribution of Tumor-Infiltrating Lymphocytes in Neuroendocrine Neoplasms

**DOI:** 10.3390/diagnostics12102340

**Published:** 2022-09-27

**Authors:** Hyung-Gyo Cho, Soo Ick Cho, Sangjoon Choi, Wonkyung Jung, Jiwon Shin, Gahee Park, Jimin Moon, Minuk Ma, Heon Song, Mohammad Mostafavi, Mingu Kang, Sergio Pereira, Kyunghyun Paeng, Donggeun Yoo, Chan-Young Ock, Seokhwi Kim

**Affiliations:** 1Department of Pediatrics, State University of New York Downstate Medical Center, New York, NY 11203, USA; 2Lunit Inc., Seoul 06241, Korea; 3Department of Pathology and Translational Genomics, Samsung Medical Center, Sungkyunkwan University School of Medicine, Seoul 06351, Korea; 4College of Pharmacy, Korea University, Sejong 30019, Korea; 5Department of Pathology, Ajou University School of Medicine, Suwon 16499, Korea

**Keywords:** neuroendocrine neoplasm, tumor-infiltrating lymphocyte, artificial intelligence, PD-L1, biomarker

## Abstract

Despite the importance of tumor-infiltrating lymphocytes (TIL) and PD-L1 expression to the immune checkpoint inhibitor (ICI) response, a comprehensive assessment of these biomarkers has not yet been conducted in neuroendocrine neoplasm (NEN). We collected 218 NENs from multiple organs, including 190 low/intermediate-grade NENs and 28 high-grade NENs. TIL distribution was derived from Lunit SCOPE IO, an artificial intelligence (AI)-powered hematoxylin and eosin (H&E) analyzer, as developed from 17,849 whole slide images. The proportion of intra-tumoral TIL-high cases was significantly higher in high-grade NEN (75.0% vs. 46.3%, *p* = 0.008). The proportion of PD-L1 combined positive score (CPS) ≥ 1 case was higher in high-grade NEN (85.7% vs. 33.2%, *p* < 0.001). The PD-L1 CPS ≥ 1 group showed higher intra-tumoral, stromal, and combined TIL densities, compared to the CPS < 1 group (7.13 vs. 2.95, *p* < 0.001; 200.9 vs. 120.5, *p* < 0.001; 86.7 vs. 56.1, *p* = 0.004). A significant correlation was observed between TIL density and PD-L1 CPS (r = 0.37, *p* < 0.001 for intra-tumoral TIL; r = 0.24, *p* = 0.002 for stromal TIL and combined TIL). AI-powered TIL analysis reveals that intra-tumoral TIL density is significantly higher in high-grade NEN, and PD-L1 CPS has a positive correlation with TIL densities, thus showing its value as predictive biomarkers for ICI response in NEN.

## 1. Introduction

Neuroendocrine neoplasms (NENs) are a heterogeneous group of epithelial neoplasms showing neuroendocrine differentiation, mostly occurring in the gastroenteropancreatic tract and bronchopulmonary tree [1,2]. According to the recent World Health Organization classification, NENs are classified into groups of neuroendocrine tumors (NET; well-differentiated, any grade) and neuroendocrine carcinomas (NEC; poor-differentiated, high-grade) [3]. Although NENs are considered rare tumors, their incidence has increased over the past two decades [4]. Generally, for patients with low-grade NENs, stable disease is the most expected overall response with the current treatment. In contrast, most patients with high-grade NENs experience disease progression, and their overall prognosis is poor, which highlights the unmet needs in NENs [2,5].

Immune checkpoint inhibitors (ICIs) targeting cytotoxic T lymphocyte-associated protein 4 (CTLA-4), programmed cell death 1 (PD-1), or its ligand (PD-L1) have shown promising treatment outcomes in various tumors by enhancing T cell immune response [6]. Likewise, recent clinical trials have observed the effectiveness of ICIs in subsets of NENs [7,8,9,10,11]. Additionally, several cases proving ICIs effectiveness in high-grade NENs were reported [12,13]. Tumor-infiltrating lymphocyte (TIL) have been recognized as a novel biomarker, besides other FDA-approved markers, because the action mechanism of ICI relies on anti-tumor immunity of TILs by inhibiting the immune checkpoints of the tumor, and the clinical outcome of ICI has correlations with PD-L1 expression and TIL across different tumor types [14,15,16,17]. Recent studies have investigated the tumor microenvironment (TME) of NENs by analyzing PD-L1 expression and TIL, revealing that, not only PD-L1 expression, but also TIL densities, are higher in high-grade NENs than in low/intermediate-grade NENs [18,19,20,21,22]. However, the results all leaned on manual quantification by pathologists, which is always at risk of inter-/intra-observer variations, and quantification was conducted with a sample size of less than one hundred [23,24].

Artificial intelligence (AI) algorithms and digital pathology-based analysis have been recently adopted to medicine, following technological advances [25,26,27,28]. AI assistance enables us to improve the accuracy of interpretation by pathologists, along with minimizing additional human technical labor and turnaround time [29,30,31]. Currently, the International Immuno-Oncology Biomarker Working Group suggested the need for computational TILs assessment, in order to reduce interobserver variability in an objective and reproducible manner [32]. We previously reported that AI-powered TIL or PD-L1 analyzer can predict clinical outcomes of ICIs in non-small cell lung cancer (NSCLC) [30,33].

Here, we evaluated PD-L1 expression and spatial TIL distributions in NENs objectively and reproducibly by using AI-powered whole slide image (WSI) analyzer in the largest sample set reported to date, consisting of greater than two hundred samples. We identified a distinctive distribution of TILs in NENs and analyzed the relationship between TIL density and PD-L1 expression to investigate the potential implication in ICI treatment.

## 2. Materials and Methods

### 2.1. Patients

Two hundred and eighteen samples of patients who were diagnosed with NENs between January 2020 and December 2021 in Ajou University Medical Center in Republic of Korea were enrolled in this study. All patient samples collected by biopsy or excised specimen were classified by the most recent World Health Organization Classification of the originated organs.

### 2.2. Tissue Image Processing

Formalin-fixed and paraffin-embedded (FFPE) tissue sections from biopsy or surgical specimens were stained with hematoxylin and eosin (H&E) and scanned by using a whole slide image (WSI) scanner, Aperio AT2 (Leica Microsystems Inc., Buffalo Grove, IL, USA), at 40× magnification (400× total magnification). NET with histological grade 1 (G1) and grade 2 (G2) were labeled as low/intermediate-grade NENs. NET with histological grade 3 (G3) and NEC with large and small cell types were labeled as high-grade NENs [34,35]. Primary origins of the NET/NEC were grouped by colorectum, stomach, small intestine, hepatopancreatobiliary tract, lung, and other organs (including anus, appendix, breast, cervix, and larynx).

### 2.3. PD-L1 Expression Assessment

FFPE tissue sections were stained with the Dako PD-L1 22C3 pharmDx kit (Agilent Technologies, Santa Clara, CA, USA). In previous studies, PD-L1 immunohistochemistry (IHC) was scored by either combined positive score (CPS) or tumor proportion score (TPS), since there is no recommended PD-L1 evaluation criteria for ICI treatment in NEN heretofore [36,37,38]. CPS represents the percentage of the number of PD-L1 positive cells, including tumor cells, lymphocytes, and macrophages, to the total number of viable tumor cells, whereas TPS only notes the proportion of PD-L1 positive tumor cells [14]. In this study, two board-certificated pathologists (S.C. and S.K.) independently interpreted the PD-L1 22C3 IHC slides and then made a consensus CPS.

### 2.4. Development of AI-Powered WSI Analyzer and Quantification of TIL Density

An AI-powered WSI analyzer, Lunit SCOPE IO, which is based on deep learning convolutional neural network model and DeepLabV3+ architecture, with a ResNet-34 backbone as feature extractor, was employed [39,40]. The AI model consists of two computer vision models: (a) cell detection model and (b) tissue segmentation model. The cell detection model locates the tumor cells and lymphocytes. The tissue segmentation model determines the tumor area, tumor stroma, or other irrelevant background regions.

Lunit SCOPE IO was developed with a 13.5 × 10^9^ μm^2^ area and 6.2 × 10^5^ TILs from 17,292 H&E-stained WSIs of multiple cancer types, including NENs [41]. The cell detection model and tissue segmentation model were developed with patches extracted from WSIs, which were annotated and segmented by 104 board-certified pathologists. For the cell detection model, 5698 and 1925 patches (1.5 × 10^5^ μm^2^ per patch) from 2485 and 849 WSIs were extracted for training and validation, respectively. The total number of annotated cells can be found in Appendix A. The performance of this model in detecting lymphocytes and tumor cells was revealed as an F1 score of 0.68 and 0.74, respectively. For the tissue segmentation model, 55,325 and 3280 patches (6.1 × 10^5^ μm^2^ per patch) from 13,958 and 849 WSIs were extracted for training and validation, respectively. The total area of annotated tissues is summarized in Appendix A. The performance of this model segmenting tumor area and tumor stroma was intersection-over-union metric, 0.82 and 0.67, respectively. 

Combining the location data with the segmentation data of lymphocytes, quantification of lymphocyte density, TIL count divided by the area of interest within the tumor area (intra-tumoral TIL density), tumor stroma (stromal TIL density), and combined tumor area-tumor stroma (combined TIL density) were produced.

### 2.5. Statistical Analysis

The Mann–Whitney U (two groups) or Kruskal–Wallis tests, followed by Tukey’s post hoc test (>two groups), were used to analyze the differences of continuous variables between groups. The Pearson correlation was used for analyzing correlations between continuous variables. *p* < 0.05 was considered a statistically significant difference. All statistical analyses were conducted using IBM SPSS Statistics, version 22.0 (IBM Inc., Armonk, NY, USA), and R, version 4.0.3, software (R Foundation for Statistical Computing, Vienna, Austria).

## 3. Results

### 3.1. Patient Characteristics

Among the samples, 158 were from colorectum, sixteen were from small intestine, fifteen were from stomach, sixteen were from hepatopancreaticobiliary tract, seven were from lung, and six were from other organs (anus, appendix, breast, cervix, and larynx). The number of cases classified as low/intermediate-grade NEN and high-grade NEN were 190 and 28, respectively. 

### 3.2. Analysis of TIL Density and PD-L1 CPS between Low/Intermediate-Grade and High-Grade NENs

The AI-powered WSI analyzer revealed that the intra-tumoral TIL density was significantly higher in high-grade NENs (11.68/mm^2^ (IQR: 4.37–31.92)), compared to low/intermediate-grade NENs (3.87/mm^2^ (1.72–9.84), *p* < 0.001) (Figure 1A)). The tendency that the high intra-tumoral TIL density was observed in the histologically high-grade NEN was identified across all types of primary organs, where NEN had originated (Appendix A). On the contrary, the stromal TIL density was not significantly different between high-grade NENs and low/intermediate-grade NENs (218.25/mm^2^ (106.55–437.10) vs. 142.45/mm^2^ (78.43–319.53), *p* = 0.179) (Figure 1B)). The combined TIL densities of high-grade NENs and low/intermediate-grade NENs also did not show statistically significant differences (86.70/mm^2^ (36.19–213.51) vs. 65.89/mm^2^ (40.14–163.04), *p* = 0.416) (Figure 1C)). The PD-L1 IHC revealed that high-grade NENs possessed a higher number of PD-L1-positive tumor cells and lymphocytes than low/intermediate-grade NENs (CPS 4.50 (2.88–12.50) vs. 0.50 (0.00–2.50), *p* < 0.001) (Figure 1D)). The tendency of showing higher CPS in the histologically high-grade NEN was identified across all types of the primary origins, except the lung (Appendix A).

We next divided the entire cases into TIL-high cases and TIL-low cases, based on the median TIL densities. The proportion of intra-tumoral TIL-high cases was greater in high-grade NEN, compared to low/intermediate-grade NEN (75.0% vs. 46.3%, *p* = 0.008; odds ratio: 3.477 (interquartile range (IQR) 1.393–9.148)). The proportion of stromal TIL-high cases and combined TIL-high cases showed insignificant differences between the two histologic subgroups. In the high-grade NEN subgroup, the proportion of cases with PD-L1 CPS ≥ 1 was 85.7%, which was significantly higher than the low/intermediate-grade NEN subgroup (33.2%) (*p* < 0.001; odds ratio 12.1 (IQR: 4.169–33.15)) (Table 1). Figure 2 shows representative case images of the intra-tumoral TIL-high NEN, intra-tumoral TIL-low NEN, stromal-TIL-high NEN, and stromal TIL-low NEN, respectively.

### 3.3. Correlation between PD-L1 CPS and TIL Density

Although the consensus on PD-L1 positivity is lacking in NEN, CPS 1 is considered a cutoff value for ICI treatment in gastric cancers, cervical cancers, and head and neck squamous cell carcinomas [42,43,44]. As we classified, regarding the cases based on the PD-L1 CPS 1, the PD-L1 CPS ≥ 1 subgroup showed significantly higher intra-tumoral, stromal, and combined TIL densities, compared to the PD-L1 CPS < 1 subgroup (intra-tumoral TIL: 7.13/mm^2^ (2.89–17.18) vs 2.95/mm^2^ (1.62–6.44), *p* < 0.001; stromal TIL: 200.9/mm^2^ (97.12–432.24) vs 120.5/mm^2^ (69.14–257.16), *p* < 0.001; combined TIL: 86.7/mm^2^ (47.06–201.31) vs 56.1/mm^2^ (33.81–110.12), *p* = 0.004) (Figure 3). Although all three classes of the TIL density revealed statistically significant differences between the CPS ≥ 1 subgroup and CPS <1 subgroup, the significance is the most evident in intra-tumoral TIL density.

A significant positive association between the log values of the PD-L1 CPS and intra-tumoral TIL density was observed (Pearson’s coefficient (*r*) = 0.37, *p* < 0.001) (Figure 4A). The positive correlations were also identified between the log value of PD-L1 CPS and both stromal TIL and combined TIL, although the correlation coefficient and degree of significance were smaller than intra-tumoral TIL (*r* = 0.24, *p* = 0.002) (Figure 4B,C). The examination of PD-L1 IHC images revealed that PD-L1-positive lymphocytes were distributed both intratumorally and peritumorally.

## 4. Discussion

This study demonstrates a distinctive distribution of intra-tumoral TIL in the TME of NEN. We identified that high-grade NENs showed significantly higher intra-tumoral TIL densities than low/intermediate-grade NENs, which were evaluated by an AI-powered TIL analyzer. In addition, the AI-analyzed TIL density, especially intra-tumoral TIL density, was positively correlated with the expression level of PD-L1. 

High-grade NENs are aggressive cancers, and the standard treatment options are limited to platinum-based regimens, such as cisplatin and etoposide [11]. In the need for a novel treatment for patients with high-grade NENs, promising evidence of the effectiveness of ICIs in high-grade NENs is evolving [45]. Yet, predictive biomarkers specific to this population are not yet established and need further investigation [11]. Prior studies have demonstrated that TMB, PD-L1 expression, and TILs were all higher in high-grade NENs, compared to low/intermediate-grade NENs, indicating that immune activation may play a significant role in the response to ICIs in NENs, and such items can be considered as potential biomarkers for predicting ICI responses in NENs [20,21,22]. 

Here, we especially focused on the spatial TIL distribution by utilizing an AI model, Lunit SCOPE IO, which enables us to standardize and reproduce the spatial TIL density in WSI. Recently, AI-based TIL evaluation has been utilized to reveal the prognostic significance in various types of cancers [46,47,48]. The AI model we utilized was previously applied to decipher the immune landscape of NSCLC and analyze the TIL distribution that correlated with the tumor response and progression-free survival of ICI in advanced NSCLC [33,49]. This study is the first to utilize an AI model to elucidate the TME in NENs. We first showed that intra-tumoral TILs were significantly enriched in high-grade NENs, and the total number of TILs (combined TIL) and stromal TILs were not different, according to the histologic grade. Recent research has revealed that tumors with a high number of intra-tumoral TILs, an “inflamed” pattern of immune phenotype, are more favorable to ICI treatment [33]. A finding that a subset of the high-grade NEN responds well to ICI can, thus, be due to their higher intra-tumoral TIL density [12,13]. Secondly, a positive linear correlation is observed between the PD-L1 CPS score and all types of TIL density. The PD-L1 CPS ≥ 1 subgroup, which is expected to be a candidate for ICI treatment, showed higher densities of not only the intra-tumoral TIL, but also the stromal TIL, as well as the combined TIL, although the levels of significance were different. As PD-L1-positive lymphocytes are distributed in both the intra-tumoral area and tumor stroma, it may be PD-L1-positive intra-tumoral TILs that provide the tumor with the immunogenicity to respond to ICI treatment.

Recent advances in AI have enabled an objective and reproducible interpretation of tasks having inter-observer variability [29,30,31]. The inter-observer variation in TIL evaluation among pathologists has long been a problem [50,51]. The accurate detection of lymphocytes in NEN is not an easy task because neuroendocrine tumor cells are typically small and uniform, thus making it difficult to distinguish from lymphoid cells, especially when the two types of cells are admixed. As shown in the representative images (Figure 2), the AI-powered analyzer could successfully discriminate lymphocytes from neuroendocrine tumor cells, even if they are admixed. In this study, the AI-powered analysis revealed an 8/mm^2^ (which corresponds to 16/10 high-power fields (HPFs)) difference in intra-tumoral TIL density between high-grade NENs and low/intermediate-grade NENs, while Busse et al. observed a 7/10 HPFs difference [21]. The difference may stem from the accuracy of TIL detection, suggesting that the AI-powered system showed higher performance in TIL interpretation than manual quantification by pathologists.

This study has some limitations. First, the sample size in high-grade NENs was relatively small (*n* = 28), compared to low/intermediate-grade NENs (*n* = 190). The small sample size of high-grade NENs impeded achieving statistically significant differences between the histological grades and TIL densities of NEN when the samples were sub-classified, according to their primary origin. Second, the subjected patients were all derived from a single institution. A multi-institutional study can expand the sample size and enables diverse analyses. Additionally, the lack of other information on the patients’ samples, such as genetic mutation profiles and transcriptomic profiles, disabled further exploration of the tumor and TME. It would be more informative to integrate the AI-powered spatial TIL distribution data into spatial genomic/transcriptomic data, which will provide comprehensive information on the TME of NENs. Lastly, the absence of clinical data hindered the evaluation of the predictive value of PD-L1 expression and TIL density in ICI for NEN patients. As a few patients had received ICI treatment to date, a multi-institutional collective study, including a sufficient number of NEN cases with matched ICI treatment response data, will enable a thorough investigation with clinical significance.

## 5. Conclusions

Together with the recent evidence of promising ICI outcomes in high-grade NENs, in our findings, the intra-tumoral TIL density and PD-L1 expression are both significantly higher in high-grade NENs, thus supporting that both the TIL density with distribution and PD-L1 expression can be evaluated as predictive biomarkers for ICI response in NENs, and further clinical investigation is warranted.

## Figures and Tables

**Figure 1 diagnostics-12-02340-f001:**
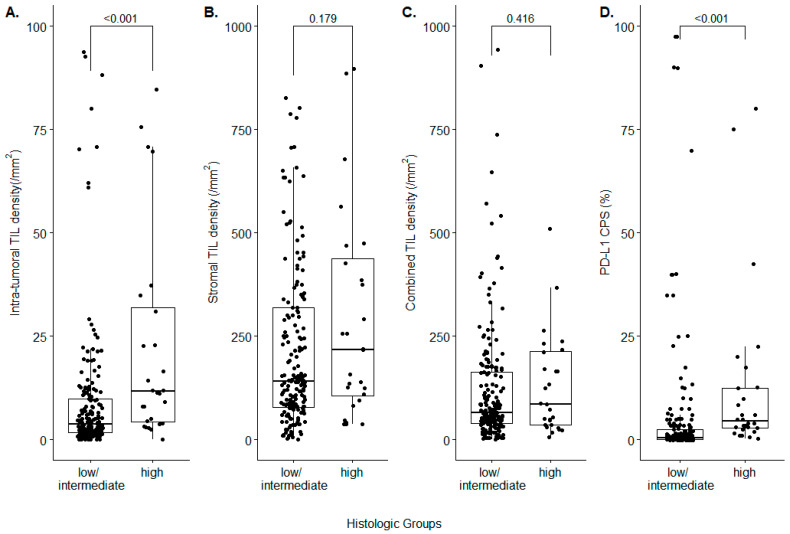
Analysis of the TIL densities and the PD-L1 expression in histologic subgroups of the neuroendocrine neoplasm. (**A**) Intra-tumoral TIL density; (**B**) stromal TIL density; (**C**) combined TIL density; (**D**) PD-L1 CPS.

**Figure 2 diagnostics-12-02340-f002:**
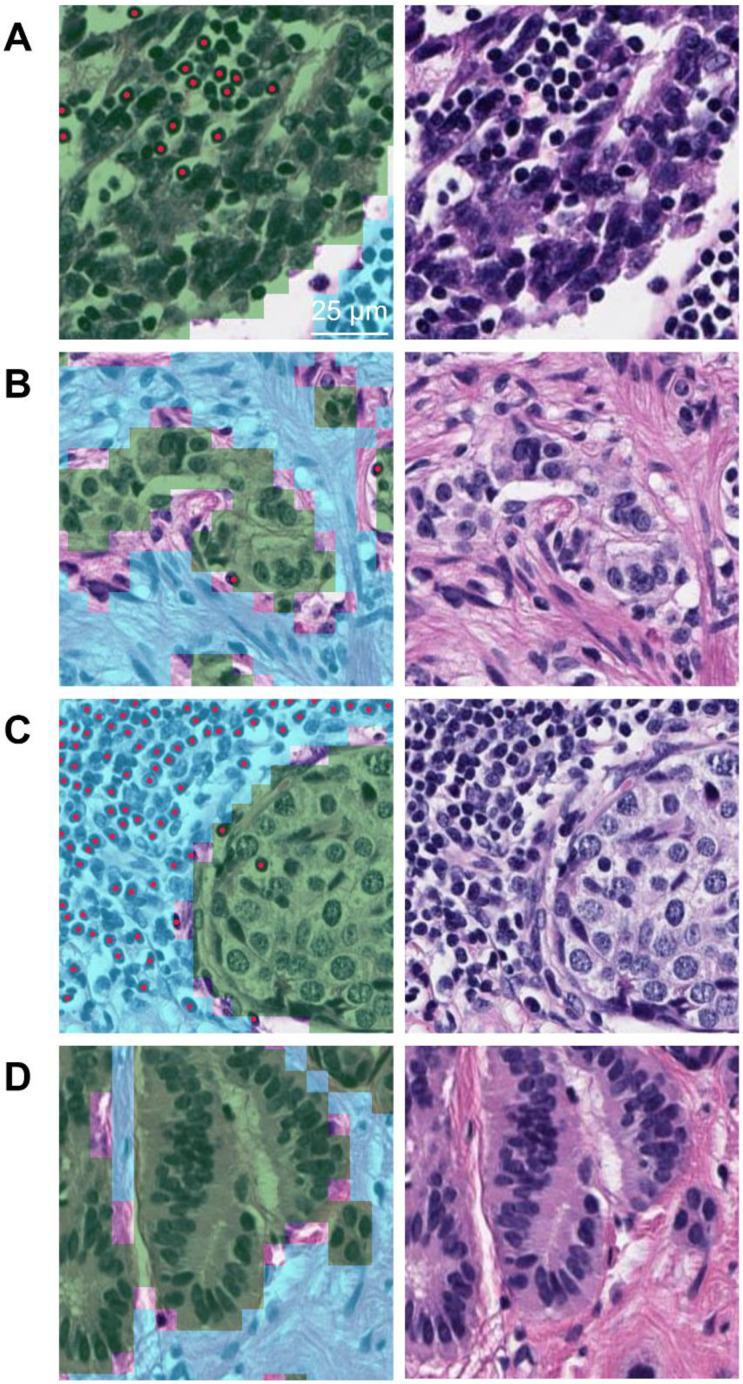
Representative images of (**A**) the intra-tumoral TIL-high NEN, (**B**) the intra-tumoral TIL-low NEN, (**C**) the stromal-TIL-high NEN, and (**D**) the stromal TIL-low NEN. Left image, H&E overlaid with AI-powered feature detection (green, tumor area; sky blue, tumor stroma; red dot, lymphocyte); right image, corresponding H&E image.

**Figure 3 diagnostics-12-02340-f003:**
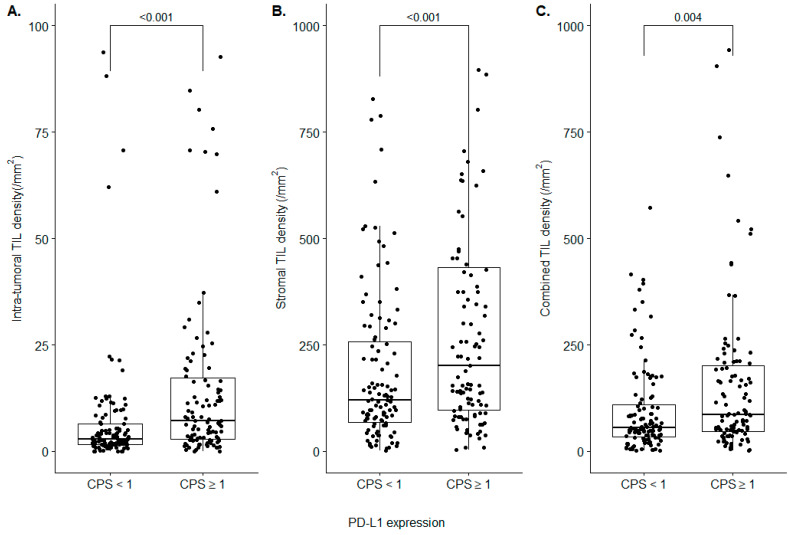
Analysis of the TIL densities in PD-L1 CPS subgroups. (**A**) Intra-tumoral TIL density; (**B**) stromal TIL density; (**C**) combined TIL density.

**Figure 4 diagnostics-12-02340-f004:**
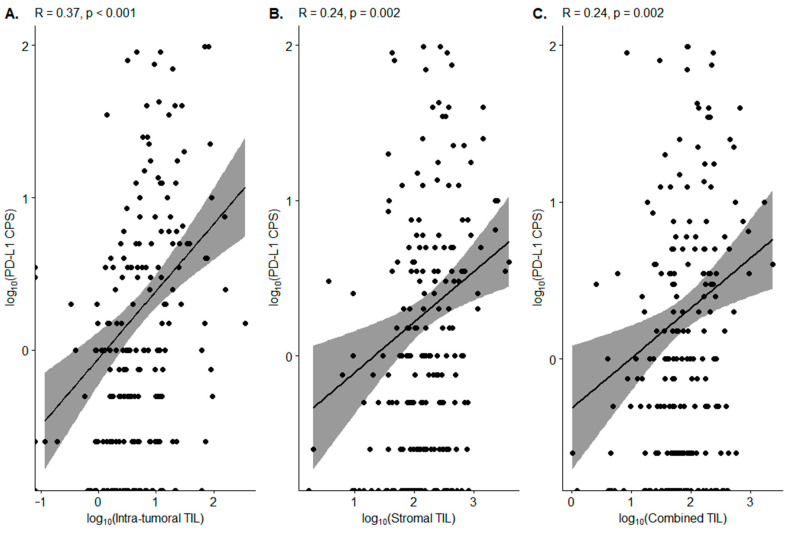
Correlation between the TIL density and the PD-L1 CPS. (**A**) Intra-tumoral TIL density; (**B**) stromal TIL density; (**C**) combined TIL density.

**Table 1 diagnostics-12-02340-t001:** Analysis of PD-L1, CPS, and TIL densities between histologic groups.

Biomarker	Group	Histologic Subgroup		*p* Value	Odds Ratio
		High-Grade NEN	Low/Intermediate-Grade NEN		
Intra-tumoral TIL	High	21/28 (75.0%)	88/190 (46.3%)	0.008	3.477 (1.393–9.148)
	Low	7/28 (25.0%)	102/190 (53.7%)		
Stromal TIL	High	16/28 (57.1%)	93/190 (48.9%)	0.544	1.391 (0.605–3.163)
	Low	12/28 (42.9%)	97/190 (51.1%)		
Combined TIL	High	16/28 (57.1%)	93/190 (48.9%)	0.544	1.391 (0.605–3.163)
	Low	12/28 (42.9%)	97/190 (51.1%)		
PD-L1 CPS	≥1	24/28 (85.7%)	63/190 (33.2%)	<0.001	12.10 (4.169–33.15)
	<1	4/28 (14.3%)	127/190 (66.8%)		

## Data Availability

The data presented in this study are available on the reasonable request to the corresponding author.

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
