# Peer review of "Artificial Intelligence-Powered Whole-Slide Image Analyzer Reveals a Distinctive Distribution of Tumor-Infiltrating Lymphocytes in Neuroendocrine Neoplasms"

_diagnostics, 2022, doi:10.3390/diagnostics12102340_

Round 1

Reviewer 1 Report

Hyung-Gyo Cho et al used artificial intelligence (AI)-powered whole-slide image analyzer to analyzed the distinctive distribution of tumor-infiltrating lymphocytes (TIL) in 218 NENs, and analyzed the correlation with proportion of PD-L1 combined positive score (CPS). Although the analysis was relatively simple, but the application of AI recognition technology on the tumor diagnosis, treatment response and even the prognosis shows promising outcome in clinical setting, thus this study has important value to reveal the application of AI in disease diagnosis filed. The methods of the manuscript are reliable, and the limitations were present in the discussion section. The English writing is good.

Author Response

We appreciate the reviewer’s comment.

Reviewer 2 Report

Major points:

I think that this manuscript seems to be relatively lengthy. The length of this manuscript should be shortened. (should be reduced by 20-25%)

1)       “Introduction” is well-written, but is too long. This section should be shortened focusing on this study.

2)      In “Materials and Methods”, the authors described as “CPS is a newly developed scoring system for PD-L1 expression …” (page 3, lines 107-). Such “new” information is unnecessary. In this section, needed “materials and methods” only should be written. Similar description is seen in other sections.

3)      In opening of “Results”, the authors described as “H&E slides obtained from total of 218 patients with NEN were included in the analysis” (page 4, lines 156-157). This overlapped with the description of “Materials and Methods” (“Two hundred and eighteen samples of patients who…) [page 2, line 86]. Similar description overlap is seen in other portions.

4)      This manuscript includes numerous abbreviations, and seems to be confusing. All these abbreviations are needed? For example, the authors described firstly in “Results” of this manuscript as “The PD-L1 immunohistochemistry (IHC) revealed that high-...” (page 4, line 174). However, “PD-L1 Expression” (page 3, line 103) itself is possible immunohistochemical expression, and the later abbreviation seems strange.

5)      Compared with the description of “Discussion”, other sections seems to be too long.

Minor points:

1.      Neuroendocrine neoplasms (NENs), rather than “Neuroendocrine Neoplasms” (page 1, line 34).

2.      Neuroendocrine neoplasms (NENs) are possibly a heterogenous group of epithelial neoplasms chiefly showing neuroendocrine differentiation; rather than “arising from neuroendocrine cells” (page 1, lines 34-35).

3.      The authors described as “their incidence (incidence of NENs) has increased over the past two decades” (page 1, lines 39-40). What is the practical frequency of the neuroendocrine neoplasms?

4.      Recurrent abbreviation: “Artificial intelligence4 (AI) algorithms …” (page 2, line 70); and “artificial intelligence (AI)-powered whole slide…” (page 2, line 79).

5.      The authors described as “in the cancer area, cancer stroma…” [page 3, line 121], “segmenting cancer area and cancer stroma” [page 3, line 137], and “within the cancer area (intra-tumoral TIL density), cancer stroma (stromal TIL density), and combined cancer area-cancer stroma (combined TIL density )…” [page 3, lines 140-142]. All NENs are “cancers”?

6.      The authors described as “the following cancer primary origins: biliary tract, breast colorectum, esophagus, head and neck, kidney, liver, lung, melanoma, ovary …. “ (page 3, lines 122-125). The “melanoma” is organ?

7.      No referred sentences. For example, the authors described “In previous studies, PD-L1 expression was scored by ….” (page 3, lines 105-107). However, there was no description of these previous reports (no references).

Author Response

We appreciate the reviewer’s comment. The revised parts in the manuscript are highlighted in yellow.

Major points:

I think that this manuscript seems to be relatively lengthy. The length of this manuscript should be shortened. (should be reduced by 20-25%)

  We have shortened the manuscript following the reviewer’s comment.

1)       “Introduction” is well-written, but is too long. This section should be shortened focusing on this study.

  We have shortened the “Introduction” section following the reviewer’s comment.

2)      In “Materials and Methods”, the authors described as “CPS is a newly developed scoring system for PD-L1 expression …” (page 3, lines 107-). Such “new” information is unnecessary. In this section, needed “materials and methods” only should be written. Similar description is seen in other sections.

  We erased the description of “newly developed scoring system…” in the manuscript. We also removed other verbose parts in the “Materials and methods” section.

3)      In opening of “Results”, the authors described as “H&E slides obtained from total of 218 patients with NEN were included in the analysis” (page 4, lines 156-157). This overlapped with the description of “Materials and Methods” (“Two hundred and eighteen samples of patients who…) [page 2, line 86]. Similar description overlap is seen in other portions.

  We deleted the redundant sentence in the first line of the “Results” section.

4)      This manuscript includes numerous abbreviations, and seems to be confusing. All these abbreviations are needed? For example, the authors described firstly in “Results” of this manuscript as “The PD-L1 immunohistochemistry (IHC) revealed that high-...” (page 4, line 174). However, “PD-L1 Expression” (page 3, line 103) itself is possible immunohistochemical expression, and the later abbreviation seems strange.

  We moved the abbreviation “immunohistochemistry (IHC)” from page 4 to page 3 in accordance with the reviewers’ comment. Also, we used the term “PD-L1 IHC” in the specific point that the PD-L1 expression was evaluated by using IHC.

5)      Compared with the description of “Discussion”, other sections seems to be too long.

 We have shortened the “Introduction,” “Materials and Methods,” and “Results” sections following the reviewer’s comment.

Minor points:

  1. Neuroendocrine neoplasms (NENs), rather than “Neuroendocrine Neoplasms” (page 1, line 34).

       We replace the term following the reviewer’s comment.

  1. Neuroendocrine neoplasms (NENs) are possibly a heterogenous group of epithelial neoplasms chiefly showing neuroendocrine differentiation; rather than “arising from neuroendocrine cells” (page 1, lines 34-35).

        We replace the term following the reviewer’s comment.

  1. The authors described as “their incidence (incidence of NENs) has increased over the past two decades” (page 1, lines 39-40). What is the practical frequency of the neuroendocrine neoplasms?

        According to the study we cited in the manuscript, which includes the data of 11 studies involving 72,048 cases, the low-grade neuroendocrine neoplasm incidence rate increased from 1.09 in 1973 to 3.51 per 100,000 in 2012. During this interval, high-grade neuroendocrine neoplasm incidence rate increased from 2.54 to 10.52 per 100,000 (Leoncini et al., Endocrine 2017).

  1. Recurrent abbreviation: “Artificial intelligence4 (AI) algorithms …” (page 2, line 70); and “artificial intelligence (AI)-powered whole slide…” (page 2, line 79).

        We erased the latter abbreviation following the reviewer’s comment.

  1. The authors described as “in the cancer area, cancer stroma…” [page 3, line 121], “segmenting cancer area and cancer stroma” [page 3, line 137], and “within the cancer area (intra-tumoral TIL density), cancer stroma (stromal TIL density), and combined cancer area-cancer stroma (combined TIL density )…” [page 3, lines 140-142]. All NENs are “cancers”?

        We replace the term “cancer area” and “cancer stroma” to “tumor area” and “tumor stroma” considering the reviewer’s comment.

  1. The authors described as “the following cancer primary origins: biliary tract, breast colorectum, esophagus, head and neck, kidney, liver, lung, melanoma, ovary …. “ (page 3, lines 122-125). The “melanoma” is organ?

        We deleted the sentence following the reviewer’s comment of shortening the manuscript.

  1. No referred sentences. For example, the authors described “In previous studies, PD-L1 expression was scored by ….” (page 3, lines 105-107). However, there was no description of these previous reports (no references).

        We inserted the citations regarding the previous studies mentioned in the sentence.

Round 2

Reviewer 2 Report

The authors sincerely responded to my comments, and the revised manuscript solved almost all my queries.